# Pathophysiological Relationship between Type 2 Diabetes Mellitus and Metabolic Dysfunction-Associated Steatotic Liver Disease: Novel Therapeutic Approaches

**DOI:** 10.3390/ijms25168731

**Published:** 2024-08-10

**Authors:** Shifat-E Ferdous, Jessica M. Ferrell

**Affiliations:** 1Department of Integrative Medical Sciences, Northeast Ohio Medical University, Rootstown, OH 44272, USA; sferdous@neomed.edu; 2School of Biomedical Sciences, Kent State University, Kent, OH 44242, USA

**Keywords:** hyperglycemia, steatosis, liver disease, metabolic disease, MASLD

## Abstract

Type 2 diabetes mellitus (T2DM), often featuring hyperglycemia or insulin resistance, is a global health concern that is increasing in prevalence in the United States and worldwide. A common complication is metabolic dysfunction-associated steatotic liver disease (MASLD), the hepatic manifestation of metabolic syndrome that is also rapidly increasing in prevalence. The majority of patients with T2DM will experience MASLD, and likewise, individuals with MASLD are at an increased risk for developing T2DM. These two disorders may act synergistically, in part due to increased lipotoxicity and inflammation within the liver, among other causes. However, the pathophysiological mechanisms by which this occurs are unclear, as is how the improvement of one disorder can ameliorate the other. This review aims to discuss the pathogenic interactions between T2D and MASLD, and will highlight novel therapeutic targets and ongoing clinical trials for the treatment of these diseases.

## 1. Introduction

Diabetes and steatotic liver disease are two of the most prominent health concerns in today’s global health landscape. Aging, sedentary lifestyle and obesity due to unhealthy dietary habits are some of the shared risk factors for developing metabolic syndrome. Current therapeutic approaches for diabetes focus on management options to maintain glucose homeostasis, whereas metabolic dysfunction-associated steatotic liver disease is an ongoing threat due to lack of treatment options. Considering the growing prevalence and overlapping pathophysiology of these two epidemics, treatment innovation is focused on common mechanistic pathways and dual-use drugs. Here, we review the interrelated pathologies of diabetes and steatotic liver disease, and present the most promising therapeutics to address these conditions. Our aim is to highlight the substantial overlap between these two conditions, with the hope it may prepare healthcare providers to pre-emptively screen and monitor patients who are vulnerable to these comorbidities.

### 1.1. Epidemiology

Diabetes mellitus is one of the fastest growing substantial global public health issues of the 21st century and is associated with many comorbidities. According to projections of the International Diabetes Federation, the global prevalence of diabetes was 537 million (10.5%) in 2021 and is likely to increase to 643 million (11.3%) in 2030 and 783 million (12.2%) in 2045. Also alarming is the growing number of people with undiagnosed type 2 diabetes mellitus (T2DM) [1].

Metabolic dysfunction-associated steatotic liver disease (MASLD), formerly known as non-alcoholic fatty liver disease (NAFLD), is a leading cause of liver-related morbidity and mortality. The approximate rates of obesity in individuals with MASLD and metabolic dysfunction-associated steatohepatitis (MASH) are 51.3% and 81.8%, respectively [2]. From a recent prospective study in individuals with diabetes, MASLD, MASH, and cirrhosis occurred in 65%, 14%, and 6% of patients, respectively [3]. The current global prevalence of MASLD is greater than 35%, and this trend is projected to increase to 55% by 2040. On the other hand, the global pooled prevalence of MASLD among patients with T2DM was 65.33%, which increased from 55.86% in 1990–2004 [4]. Despite these facts, it is still wholly unclear if MASLD is either the origin or outcome of T2DM.

The risk of developing T2DM is more than two times higher in patients with MASLD compared to patients with normal hepatic function [5]. This incidence also rises with MASLD severity. A retrospective study with biopsy-confirmed MASLD patients without initial T2DM reported that a significantly higher proportion of patients with fibrosis stages 3–4 developed T2DM (51%) than those with fibrosis stages 0–2 (31%) over a mean follow-up of 18.4 years [6].

### 1.2. Type 2 Diabetes Mellitus (T2DM)

T2DM is a chronic metabolic disorder characterized by inadequate control of blood glucose, which usually occurs when pancreatic β-cells cannot produce enough insulin, or the body cannot effectively use the insulin produced by these cells. Chronic hyperglycemia is associated with both oxidative stress and inflammation in the liver and other tissues, and complications from T2DM represent significant contributors to human morbidity. The prevention and control of T2DM through medications along with lifestyle modifications remains an ongoing challenge and the complexity of T2DM has raised consequential interests in developing newer pharmacotherapeutic approaches worldwide.

### 1.3. Metabolic Dysfunction-Associated Steatotic Liver Disease (MASLD)

MASLD is a clinicopathological condition stemming from excessive triglyceride (TG) accumulation in hepatocytes and gives rise to a spectrum of liver complications from simple steatosis to hepatocellular carcinoma (Figure 1). MASLD affects around one in every four individuals globally [3], and is becoming a significant cause of hepatocellular carcinoma and liver transplantation. MASLD is also prevalent in children and its current frequency is approximately 7% among the general pediatric population. This distressing condition is rising synchronously with the global epidemics of obesity and T2DM [7]. Poor diet, sedentary lifestyle, increasing age and obesity (BMI ≥ 30) are among the other factors driving the increasing trend of MASLD [8]. Over 40% of reported MASLD cases are detected in individuals above 60 years of age, and older patients with MASLD are more prone to developing fibrosis and cirrhosis [9]. MASLD progression to more serious liver disease occurs from additional metabolic stimuli, including inflammatory insults, oxidative stress, lipotoxicity and/or dysbiosis that results in MASH. Left untreated, MASH may develop into cirrhosis. While sometimes reversible [10], cirrhosis may further advance to hepatocellular carcinoma in a subset of patients (Figure 1).

Obesity and insulin resistance (IR) due to T2DM are two predominant factors that may escalate MASLD. A variety of both monotherapies and combinations of oral and injectable drugs have been approved by the United States Food and Drug Administration (USFDA) to treat T2DM. The very recently USFDA approved (March 2024) drug Resmetirom is the only treatment option available for MASLD and MASH, though only 25% of patients saw improvement of fibrosis by one score with no worsening of fibrosis [11]. The overlapping correlation between these two metabolic disorders spotlights the necessity of discovering new preventive and therapeutic measures to tackle these substantial clinical burdens. This review will focus on the current therapeutic approaches for T2DM and outline some promising future novel pharmacological targets for T2DM and MASLD by considering their distinctive molecular pathologies.

### 1.4. Pathophysiological Relationships between T2DM and MASLD

Obesity often leads to the development of T2DM, and these conditions are significant risk factors for developing MASLD. With obesity, imbalance between daily energy intake and expenditure leads to excess adipose tissue formation. IR and inflammation contribute to morbidity in patients with obesity, and inflammation is also considered to play a crucial role in the development of IR [12]. IR within T2DM is a major contributing phenotype to the development of MASH. In the liver, insulin receptor signaling triggers insulin receptor substrate (IRS) and phosphoinositide-3-kinase (PI3K) to activate serine/threonine kinase (Akt). Akt can then (1) increase glycogen production by inhibiting the phosphorylation of glycogen synthase kinase 3β, and (2) suppresses gluconeogenesis by inhibiting forkhead box O1, a key regulator of gluconeogenetic genes. In the presence of mitochondrial dysfunction, oxidative stress and inflammation, the serine phosphorylation of IRS protein leads to the downregulation of IRS/PI3K/Akt and results in IR [13,14]. IR decreases glycogen synthesis and increases hepatic gluconeogenesis and glycolysis, which can further trigger the progression of T2DM. The following mechanisms contribute to liver dysfunction in the presence of T2DM: (a) poor glycemic control due to increased hepatic gluconeogenesis and glycolysis further provides substrates for de novo lipogenesis and thereby results in hepatic steatosis [15]. It has also been shown that increased glycated hemoglobin (HbA1c) levels (>7.0%) due to poor glycemic control is linearly associated with the progression of hepatocyte ballooning (swelled, rounded and deformed hepatocyte structure due to fat droplets accumulation and stress-induced endoplasmic reticulum expansion) and fibrosis severity. The study demonstrated that for every 1% rise in HbA1c level, the odds of advancing to a higher fibrosis stage increase by 15% [16]. Other IR-mediated molecular mechanisms include: (b) hepatic mitochondrial dysfunction due to oxidative stress-induced release of inflammatory mediators (IL-6, IL-1β, TNF-α), followed by c-Jun N terminal kinase signaling-mediated cell damage, (c) obesity and hyperlipidemia due to imbalance in adipokines from adipose tissue that results in lipotoxicity and collagen deposition, and (d) gut microbiota dysbiosis due to a chronic high-fat diet that produces metabolic ethanol, alters gut permeability and increases endotoxin release to activate toll-like receptor inflammatory pathways in the liver [17,18] (Figure 2A–D).

IR-mediated lipotoxicity due to poor metabolic control is another predominant factor in MASLD development in patients with T2DM. In a state of IR, adipose tissue lipolysis increases due to decreased adiponectin-mediated free fatty acid (FFA) accumulation, resulting in a large amount of FFAs which are effluxed to hepatocytes. In healthy hepatocytes, FFAs usually generate fatty acyl-CoA, which enters the mitochondrial β-oxidation cycle to produce energy. With excess FFA influx, normal respiratory oxidation processes are disrupted, leading to impaired lipid oxidation along with increased toxic metabolites and reactive oxygen species (ROS). Lipotoxicity then promotes ROS-mediated inflammatory responses, resulting in steatosis progression toward fibrosis [19].

Chronic excess calorie intake from a high-fat diet causes ER stress by promoting de novo lipogenesis and induces inflammatory responses. The consequence is hepatic insulin resistance and elevated glucose production via the activation of gluconeogenesis. Furthermore, disrupted lipid metabolism in addition to gut microbiome-derived endotoxins stimulates the production and release of proinflammatory cytokines (IL-1, IL-6, and TNF-α) which contributes to the worsening of IR in T2DM [18].

Research is also focused on finding a relationship between bile acid metabolism and T2DM. The bile acid pool is composed of primary bile acids synthesized in the liver which are then converted to secondary bile acids in the enterocytes by gut microbiota. It has been reported that IR is associated with the increase of some of the primary, secondary and/or conjugated bile acids [20,21]. Bile acid homeostasis is modulated by a fasting or feeding state. A high-fat diet disrupts this homeostasis and leads to the pathogenesis of obesity and IR. In the postprandial state, bile acids activate farnesoid X receptor (FXR) to inhibit hepatic lipogenesis by stimulating IRS/PI3K/Akt pathway, which inhibits the mechanistic target of rapamycin complex 1 (mTORC1) and induces autophagy. mTORC1-pS6K promotes maturation and nuclear localization of steroid regulatory element binding protein 1c (SREBP1c) to stimulate lipogenesis [22].

During the early stage of fasting, the body secretes less insulin due to reduced glucose, and glucagon is increased to induce hepatic glycogenolysis, thus reducing the risk of hypoglycemia. As fasting is prolonged and hepatic glycogen is depleted, gluconeogenesis and ketogenesis become the major energy sources. Glucagon also stimulates the release of lipase-A enzymes to initiate the breakdown of triglycerides to free fatty acids, which activate peroxisome proliferator-activated receptor α (PPARα) in hepatocytes and PPARγ in adipose tissue to induce fibroblast growth factor 21 (FGF21). FGF21 induces peroxisome proliferator-activated receptor-γ coactivator 1α (PGC-1α) to stimulate mitochondrial oxidative phosphorylation and energy production. FGF21 also prevents mTORC1 signaling to upregulate insulin signaling [23,24,25].

## 2. Novel Drug Approaches Common in T2DM and MASLD

### 2.1. GPR119 Agonists and Incretin Mimetics

GPR119 is a stimulatory G-protein-coupled receptor expressed in pancreatic β-cells, enteroendocrine K- and L-cells, brain, liver and other tissues [26]. The activation of GPR119 induces the release of the incretin hormones GLP-1 and GIP from L- and K-cells, respectively, which triggers insulin release from pancreatic β-cells and directly induces insulin secretion via GPR119 receptors in the pancreas. GPR119 agonism in the liver activates AMP-activated protein kinase (AMPK) by promoting phosphorylation. The activated AMPK decreases de novo lipogenesis by inhibiting the lipogenic enzymes ACC, FAS and SCD1 via the inhibition of SREBP1c [27]. DA-1241, a GPR119 agonist by Neurobo Pharmaceuticals, is being studied for liver enzyme and glucose outcomes in healthy volunteers and patients with T2DM in a Phase 1 study (NCT03646721) and is now under ongoing a clinical Phase 2 trial for subjects with presumed MASH (NCT06054815). Another agent, MBX-2982, also showed a positive outcome in reducing postprandial glucose in a 4-week Phase 2 clinical trial in patients with T2DM (NCT01035879).

Incretin mimetic drugs (GLP-1 agonists and GIP agonists) are already approved by the USFDA for reducing IR in T2DM and are now under trials to observe their efficacy in MASLD. The FDA-approved anti-diabetic drug Tirzepatide (2022) by Lilly reduced liver fat content in patients with T2DM (NCT03882970), which points to its potentiality to treat MASH too. A very recently completed trial (NCT04166773) evaluated tirzepatide for the resolution of MASH after 52 weeks of weekly injections. MASH resolution was achieved in 73.9% of patients receiving the highest dose [28]. Pemvidutide is another novel investigational GLP-1/glucagon dual agonist for MASH for which the FDA granted fast-track approval as a MASH therapy considering its efficacy (68.5% liver fat reduction) achievement within just 12 weeks (NCT05006885). GLP-1 receptor agonists were also shown to reduce visceral adipose, waist circumference, total cholesterol and serum triglycerides in patients with both T2DM and MASLD [29]. The mechanisms by which GLP-1 improves liver function are not known, but may involve AMPK-mediated activation of PPARα (below), suggesting that common pathways may be involved [30].

### 2.2. Peroxisome Proliferator-Activated Receptor (PPAR) Agonists

PPARs are one of the beneficial regulators of metabolic pathways, which make them promising drug targets for both T2DM and MASLD. PPARs are nuclear receptors having three isoforms based on their tissue distributions and functions: PPARα, PPARβ/δ and PPARγ. PPARα exerts its main action in the liver and skeletal muscle where it transcriptionally regulates the genes involved in gluconeogenesis, β-oxidation, ketogenesis, fatty acid uptake and lipid transport. PPARγ is highly expressed in white and brown adipose tissue and promotes adipogenesis, insulin sensitivity and lipogenesis, and upregulates adiponectin. Adiponectin is necessary for AMPK activation which is required for increasing glucose uptake and fatty acid oxidation [31]. PPARβ/δ is found in different tissues including the liver, skeletal muscle and brown adipose tissue, and it also enhances fatty acid metabolism, as well as preserves glucose [32]. Thiazolidinediones (Rosiglitazone, Pioglitazone) are an established class of PPARγ agonists for hyperglycemia. From previous trials with other selective PPAR agonists (PPARα agonist: fibrates; PPARδ agonist: seladelpar), it was concluded that they have a potential role in the improvement of MASLD [33]. Current research focus is shifting toward developing dual or pan-PPAR (action on all three isoforms) agonists to simultaneously target different organ-specific MASLD pathways, including Elafibranor (PPARα/δ agonist), saroglitazar (PPARα/γ agonist), and lanifibranor (PPAR α/γ/δ agonist). Elafibranor is currently not under trial continuation because its Phase 3 study did not meet the predefined primary efficacy endpoint, though no safety issues were identified (NCT02704403). Saroglitazar underwent Phase 2 trials NCT03061721 (16 weeks, completed [34]), NCT03863574 (24 weeks, completed, no result posted), NCT03639623 (24 weeks, completed [35]) and is recruiting for several more (NCT05011305, NCT05872269). Likewise, lanifibranor underwent a Phase 2 trial NCT03459079 (24 weeks, completed) and is recruiting for a Phase 3 trial (NCT04849728) for the treatment of MASLD.

### 2.3. THR Agonists

Thyroid hormone receptors (THR) are nuclear receptors that are crucial for glucose homeostasis, as well as lipid metabolism. Between the two isoforms of this receptor, THR-β expression in liver tissues is more highly expressed than THR-α [36]. Thyroxin (T4) is converted to T3 (active form) in response to increased thyroid stimulating hormone (TSH) from the pituitary gland. Both MASLD and T2DM are induced by hypothyroidism, which typically involves increased secretion of TSH but reduced circulating T3 and T4Agonists activate THR target genes, which results in increased pancreatic insulin sensitivity and reduced hepatic TG accumulation [37,38]. As mentioned earlier, the recently approved drug Resmetirom acts on hepatic THR-β. Resmetirom showed a significant reduction in liver fat (~33%) in 12 weeks, as well as MASH resolution without the worsening of fibrosis (~25%) within 36 weeks of treatment in patients with MASLD and fibrosis [39]. Based on the promising results from Phase 2 trials, this drug entered in a Phase 3 MAESTRO clinical trial program to evaluate its safety and tolerability along with efficacy endpoints within a large scale of patient groups. Moreover, it was also observed that Resmetirom seems to affect the conversion of T4 to T3 (T4 level decreased ~17–21%; as well as TSH level; T3 level unchanged) [11], which makes this drug possibly effective in increasing insulin sensitivity. Another THR-β agonist, VK2809, is selective for hepatocytes and results are awaited from a clinical trial in patients with hypercholesterolemia and MASLD (NCT02927184). Another ongoing trial of VK2809 is investigating the reduction in steatosis after 52 weeks of treatment in patients with MASLD and fibrosis (NCT04173065). TERN-501 is another THR-β agonist which has completed a Phase 2a trial (NCT05415722) for steatosis, either alone or in combination with TERN-101 (an FXR agonist). On the other hand, exenatide, a GLP-1 receptor agonist, showed improved pituitary and hypothalamic sensitivity to thyroid hormone action in patients with T2DM by reducing serum TSH levels in 12 months [40].

### 2.4. SGLT2 Inhibitors

Sodium–glucose cotransporter 2 (SGLT2) inhibitors are anti-diabetic drugs that inhibit renal glucose resorption and promote urinary glucose elimination. Several recent studies have demonstrated that SGLT2 inhibitors exhibit anti-fibrotic effects and reduce liver fat. In a trial of more than 7000 patients with T2DM, the SGLT2 inhibitor empagliflozin significantly reduced ALT and AST, though liver fat and fibrosis were not measured outcomes [41]. Another systematic review confirmed these findings and found that SGLT2 inhibitors also reduced liver fat and visceral fat mass in patients with T2DM and MASLD [42]. Empagligflozin may exert anti-inflammatory effects while simultaneously activating autophagy in hepatic macrophages [43], while another study demonstrated the alleviation of fibrotic pathways in hepatic stellate cells [44]. Importantly, studies suggest that these effects may be long-lasting, as liver fibrosis remained significantly reduced from baseline after 3 years [45] and 5 years [46,47] in patients with T2DM and MASLD who take SGLT2 inhibitors.

### 2.5. Drugs Targeting the Mitochondria

#### 2.5.1. Mitochondrial Pyruvate Carrier (MPC) Modulator

Mitochondrial dysfunction plays one of the prime roles in MASLD and diabetes pathogenesis. Pyruvate is the product of glucose metabolism through glycolysis. After being produced in the cytosol, pyruvate crosses mitochondrial pyruvate carrier (MPC) located in the mitochondrial inner membrane and stimulates citric acid cycle and oxidative phosphorylation to produce malate and citrate. MPC dysregulation can lead to diverse pathologic conditions. Hepatic tissue-specific MPC inhibition can attenuate hyperglycemia and fat accumulation by preventing malate-induced gluconeogenesis and citrate-induced de novo lipogenesis, respectively [48]. MSDC-0602K is a potential MPC modulator which increases insulin sensitivity by directly targeting MPC in liver. In a completed 52-week double-blind Phase 2b study (NCT02784444), MSDC-0602K reduced blood glucose, HbA1c, insulin and liver enzymes compared to the placebo [49]. Another new study (NCT03970031) is planned to investigate the improvement in glycemic control and cardiovascular outcomes with MSDC-0602K in patients with pre-T2DM or T2DM and MASLD/MASH.

#### 2.5.2. Mitochondrial Uncoupling Agents

Mitochondrial uncoupling is a physiological process during the final step of oxidative phosphorylation in which protons normally pumped to the inner mitochondrial membrane are shifted by uncoupling proteins (UCP) to increase energy expenditure and thermogenesis. This UCP-led backflow of protons from the inner membrane to the mitochondrial matrix leads to thermogenesis instead of ATP synthesis. Increased hepatic thermogenesis and energy expenditure results in increased hepatic insulin sensitivity and glycogen synthesis, along with reduced very low density lipoprotein, improving glycemic and hepatic steatotic conditions [50]. A Phase 2a trial on HU6 (NCT04874233), a novel mitochondrial uncoupler, demonstrated approximately 30% liver fat reduction in overweight patients with MASLD [51]. Another proposed trial (NCT06104358) is currently recruiting to investigate HU6-mediated energy metabolism in patients with obesity and T2DM.

### 2.6. Direct AMPK Activators

AMPK is a serine/threonine kinase which acts as a primary regulator of cellular metabolism. During a hypophagic or nutrient-deficient phase, the activation of AMPK triggers catabolism (glucose uptake and lipid oxidation) and inhibits anabolism (glucose and lipid production) to maintain energy balance. PXL-770 is a direct AMPK activator which showed promise for improvement of T2DM and steatosis. While PXL-770 reduced hyperglycemia and enhanced insulin sensitivity, it did not meet the primary endpoint of reduced liver fat (NCT03763877 [52]). A-592107 and A-769662 are other small-molecule direct AMPK activators currently under pre-clinical or in vitro studies [53] (Figure 3).

## 3. Treatment Approaches for T2DM

Current pharmaceutical treatment approaches target the maintenance of glycemic control as there is no curable treatment for T2DM. Table 1 summarizes the current anti-diabetic drug classes.

### 3.1. Novel Treatment Strategies for T2DM

In the past few years, T2DM management has significantly expanded and with the emergence of the betterment in patient care, significant novel treatment approaches have been identified for early commercialization. Among these strategies, receptors and enzymes mediating glucose homeostasis are at the frontline of T2DM therapeutic target identification.

#### 3.1.1. SNO-CoA-Assisted Nitrosylase (SCAN) Enzyme Inhibition

Coenzyme A (CoA) is a cofactor for several cellular processes, notably fatty acid oxidation and the citric acid cycle. It was recently discovered that CoA can carry endogenous nitric oxide in the form of S-nitrosothiol (SNO-CoA) [54], which is catalyzed by SNO-CoA-assisted nitrosylase (SCNA). Hypernitrosylation of the insulin receptor and IRS1 results in IR [55]. In pre-clinical studies, it was shown that mice with SCAN knockdown were protected from IR mediated T2DM [55]. This novel discovery may lead to further study into SCAN as a therapeutic target for T2DM.

#### 3.1.2. Ketohexokinase Inhibition

Ketohexokinase (KHK) phosphorylates fructose and commits its metabolic fate in the liver. Human patients with mutations in KHK fail to metabolize fructose and instead exhibit accumulation in blood and urine. The knockdown of KHK reduced weight gain and metabolic dysfunction in genetically obese mice and conversely, KHK gene expression was positively correlated with MASH in children [56]. Therefore, inhibition of KHK is an attractive therapeutic target for patients with both T2DM and MASLD, as KHK inhibition results in reduction in glucose resynthesizing from the fructolysis metabolites as well as lipogenesis reduction [57]. A randomized controlled trial performed in patients with MASLD and T2DM found that KHK inhibition by PF-06835919 resulted in a significant reduction in liver fat and modest but non-significant reductions in HbA1c (NCT03969719) [58]. Additional trials (NCT04193436, NCT03256526) [59] were completed to examine the efficacy of PF-06835919 in reducing steatosis and metabolic dysfunction associated with MASLD and T2DM.

#### 3.1.3. Angiopoietin-Related Protein-3 (ANGPTL3) Inhibition

Angiopoietin-related proteins are important regulators of lipoprotein metabolism. Angiopoietin-like protein-3 (ANGPTL3) is secreted from the liver and reduces plasma TG levels by inhibiting lipoprotein lipase (LPL). Increased ANGPTL3 was previously reported in individuals with diabetes as well as patients with obesity but without T2DM [60]. ANGPTL3-induced stagnation of carbohydrate metabolism can be augmented with IR through increased lipolysis and can lower insulin sensitivity by the attenuating de novo lipogenesis in white adipose tissue [61]. ISIS-703802 (Vupanorsen) is an ANGPTL3 inhibitor with proven dose-dependent TG level lowering efficacy of up to 53% in patients with T2DM (NCT03371355) [62].

#### 3.1.4. 11-β Hydroxysteroid Dehydrogenase-1 (11β-HSD1) Inhibition

11β-hydroxysteroid dehydrogenase 1 (HSD1) metabolizes cortisone to its active form of cortisol and eventually other glucocorticoids. 11β-HSD1 is highly expressed in the liver and adipose tissue. One of the most prominent functions of cortisol is to activate hepatic gluconeogenesis regulatory enzymes by inducing the transcription of phosphoenolpyruvate carboxykinase and glucose-6-phosphatase. On the other hand, in adipose tissue cortisol enhances lipolysis and leads to glycerol release, which can also be used as a substrate for gluconeogenesis. 11β-HSD1 inhibitors prevent cortisol-induced upregulation of glucose [63,64] and lowered plasma glucose inadequately controlled by metformin monotherapy (NCT00698230) [65].

#### 3.1.5. Lyn Protein Tyrosine Kinase Activation

Lyn is a member of the Src family of non-receptor protein tyrosine kinases. Activation of lyn has been found to effectively improve glucose homeostasis, in part due to its rapid activation. By increasing phosphorylation of IRS1, it induces translocation of insulin-sensitive glucose transporters to permit more glucose entry into, thus decreasing circulating blood glucose level [66]. MLR-1023, a novel lyn protein tyrosine kinase agonist which works as a non-PPARγ insulin sensitizer to treat uncontrolled T2DM, showed proven efficiency in reducing fasting plasma glucose up to 39.3 mg/dl from baseline in a Phase 2 study (NCT02317796) [67].

## 4. Current Treatment Approaches for MASLD

In 2024 the first-ever and only MASH treatment drug, Resmetirom (under the brand name Rezdiffra by Madrigal Pharmaceuticals) was approved by the USFDA to treat liver fibrosis and also steatohepatitis. Prior to this, lifestyle modification through exercise, weight loss and dietary changes have been recommended for the management of MASLD and MASH fibrosis. Resmetirom is a partial agonist of the thyroid hormone beta (THR-β) receptor and stimulates this abundant receptor in the liver to reduce intrahepatic triglycerides [68]. One major advantage of this drug is its specificity in targeting THR-β which reduces off-target side effects typically seen with THR-α drugs. Still, only about 25% of patients receiving resmetirom responded with improvement of fibrosis without worsening of MASH [11], indicating there is still an unmet need for additional therapeutic options.

One such approach is through the investigation of established drugs with off-label benefits for MASLD. Many of these drugs are already approved for hyperglycemia, obesity or hypertriglyceridemia. Clinical trials are being conducted to determine if these drugs can be utilized for the management of MASLD, in part due to the pathophysiological correlation of these conditions with MASLD progression [69], and select drugs are listed in Table 2.

### 4.1. Novel Bile-Acid-Based Treatment Approaches for MASLD

Finding suitable treatment options to treat steatotic liver diseases effectively is an active area of research. As of May 2024, a total of 1394 MASLD trials were registered with clinicaltrials.gov (accessed on 8 August 2024); among these are Phase 1: 166, Phase 2: 331, Phase 3: 81 and Phase 4: 76, and others are trials of behavioral interventions. Some of the most promising emerging molecular targets for the treatment of MASLD are summarized below and in Figure 4.

#### 4.1.1. Farnesoid X Receptor (FXR) Agonism

Farnesoid X receptor (FXR) is a hepatic nuclear receptor that plays a key role in the regulation of bile acid synthesis. Additionally, the activation of endogenous FXR by bile acids promotes glucose use in the liver and reduces steatosis; therefore, FXR agonists are promising drug targets for MASLD. Under the class of steroidal FXR agonists, obeticholic acid (OCA), a semi-synthetic bile acid derivative which is already approved to treat primary biliary cholangitis (PBC), and EDP-305 are now undergoing clinical trials for MASH. Two Phase 2 trials of OCA (NCT01265498, NCT00501592) demonstrated MASLD score improvement with reduced ALT, AST and GGT levels, as well as dose-dependent increased insulin sensitivity in patients with T2DM and MASLD [70,71]. The Phase 3 REGENERATE trial (NCT02548351) revealed OCA dose-dependently reduced serum hepatic biomarkers in patients with pre-cirrhotic fibrosis [72]. A Phase 2 trial of the oral FXR agonist EDP-305 (NCT03421431) also demonstrated dose-dependent reductions in ALT, liver fat percentage, and other MASH and fibrosis markers [73]. However, due to poor pharmacokinetic properties and pruritis side effects non-steroidal FXR agonists are also being explored, including cilofexor, tropifexor and nidufexor [74].

#### 4.1.2. Takeda G-Protein-Coupled Receptor (TGR5) Agonism

Takeda G-protein-coupled receptor 5 (TGR5, also known as G-protein-coupled bile acid receptor/GPBAR1) is expressed mainly in the intestine, skeletal muscle, brain, adipose tissue and non-parenchymal cells in the liver. Similar to FXR, TGR5 binds endogenous bile acids to exert beneficial effects related to glucose and energy metabolism. Activation of TGR5 induces GLP-1-mediated increases in glucose metabolism with resultant increases in hepatic glycogenesis and insulin secretion from pancreatic β-cells, and it inhibits proinflammatory cytokines through NF-κB inhibitory signaling in Kupffer cells [75]. INT-777 is a synthetic TGR5 agonist that improved glycemic control and reduced steatosis and weight gain in high fat diet-fed mice [76]. Another TGR5 agonist, BAR501, improved energy homeostasis by inducing adipose tissue browning in steatotic mice [77,78]. RDX8940 is an intestine-specific TGR5 agonist that also showed potency in improving liver steatosis and insulin sensitivity in a MASLD mouse model [79]. Despite these positive results, limited trials have studied the activation of TGR5 in humans. One such study (NCT01674946) determined that bile acid-based improvement in blood glucose via GLP-1 may actually involve FXR-dependent pathways [80].

As both FXR and TGR5 are involved in improving steatosis and glucose metabolism, dual agonists are also investigated to check their complementary and synergistic effects in these metabolic pathways. INT-767 is a steroidal FXR-TGR5 dual agonist that demonstrated anti-fibrotic effects in pre-clinical trials [81,82]. BAR502, another non-steroidal FXR-TGR5 dual agonist, also showed thermogenetic efficacy by adipose tissue browning [83], and histopathological improvement in combination with UDCA in pre-clinical studies [84].

#### 4.1.3. Fibroblast Growth Factor (FGF) Analogs

FGF15 (mouse)/FGF19 (human) and FGF21 are both under the FGF19 family of endocrine fibroblast growth factor. FGF19 is expressed in enterocytes whereas FGF21 is highly expressed in hepatocytes (Figure 3). With the presence of non-signaling transmembrane co-receptor β-klotho, FGF19 suppresses bile acid synthesis, while FGF21 inhibits lipolysis and preference for carbohydrate consumption [85]. The FGF19 analog Aldafermin, after completing Phase 1 (NCT04823702) and Phase 2 ALPINE trials (NCT02443116, NCT03912532, and NCT04210245), showed efficacy in the reduction of hepatic fat in patients with MASH [86,87].

The FGF21 analog Pegbelfermin (BMS-986036) has completed two Phase 2 trials (NCT03486899 and NCT03486912) to evaluate its efficacy in stage 3 fibrosis and cirrhosis, respectively [88,89]. Another long-acting subcutaneous FGF21 analog, B1344, is currently recruiting for a Phase 1 trial (NCT05655221) after its pre-clinical efficacy validation [90]. Efruxifermin and Pegozafermin are two other FGF21 analogs with completed Phase 2 trials (NCT03976401 and NCT04048135, respectively) for MASH [91]. These drugs are also in trials currently recruiting to evaluate their efficacies in MASH with fibrosis and cirrhosis.

Several other non-bile acid-based therapeutic targets are currently being investigated in various trials and are summarized in Table 3.

## 5. Discussion

T2DM and MASLD candidate drugs under pre-clinical or clinical development stages are paving the way to overcome the treatment obstacles of these two clinical challenges. Significant progress has been made in antidiabetic therapeutics in the last decade with the development of several dual and triple combination agents with unique formulations. Research is still ongoing to uncover the detailed molecular mechanisms that contribute to this condition and its relationship to MASLD.

Until recently, patients with MASLD were totally dependent on disease management by lifestyle modification. As MASLD is a multifactorial disease, the current pipeline treatment options with specified targets may not be sufficient to thoroughly overcome this disease. Instead, combinations of drugs targeting multiple molecular pathways may exhibit more promising and desired therapeutic potential, though precision therapy based on individual MASLD phenotype may still be required. It is very likely that patients with MASLD might have varied pathophysiological alterations which cannot be counteracted with a specific targeted mechanism. As T2DM and MASLD share multiple pathophysiological interrelationships (insulin resistance, systemic inflammation, oxidative stress and lipotoxicity), drugs that target both the metabolic and inflammatory pathways could offer combined benefits to manage both these diseases. Reducing triglyceride synthesis by increasing adiponectin activity with higher insulin sensitivity is one prospective pathway. Research efforts in the next decade should focus on identifying promising drugs that target the interplay between these and other pathways. For treating the initial stages of MASLD, we believe that addressing the metabolic risk factors related to triglyceride-adiponectin balance, and glucose and lipid metabolism should be kept in mind, whereas liver-specific drug targets to reverse inflammation and collagen deposition become more crucial when the disease progresses to advanced fibrosis and cirrhosis. Currently, active research is focused on bile acid signaling as a major regulatory pathway for both glucose and lipid metabolism, which can serve as a unique target for both MASLD and T2DM therapy. From this viewpoint, FXR-TGR5 dual agonists can be potentially control both diseases by lowering triglycerides [92], reducing ChREBP [93], improving insulin sensitivity via FXR [94] as well as GLP-1 [76], and exerting potent anti-inflammatory effects via TGR5 [79]. Other novel dual-acting medications and lipid metabolism modulators, for example, the enzymatic inhibitors of de novo lipogenesis that target ACC [95], SCD1 [96], or DGAT2 [97], as well as PPAR agonists [33] and antioxidants (e.g., vitamin-E) [98] that reduce disease severity, are other potential approaches which may lead to shared beneficial outcomes to combat these increasingly prevalent and inter-related diseases.

## 6. Conclusions

T2DM and MASLD are some of the most prevalent diseases worldwide and share several common pathological pathways. MASLD is the most common liver disease in patients with T2DM and patients with MASLD have a two-fold risk of developing T2DM [99]. Therefore, targeting either T2DM or MASLD may exert beneficial effects for both conditions, and current research is pushing for drugs that target both pathways. De novo lipogenesis, insulin resistance, lipotoxicity, and gut dysbiosis are responses to chronic metabolic stress that promote disease progression and represent therapeutic targets for upcoming research. The recognition of the significant overlap between T2DM and MASLD will (1) highlight opportunity for novel therapeutic targets with dual benefits, and (2) provide clinicians with additional insight into the pathological crosstalk that contributes to each condition.

## Figures and Tables

**Figure 1 ijms-25-08731-f001:**
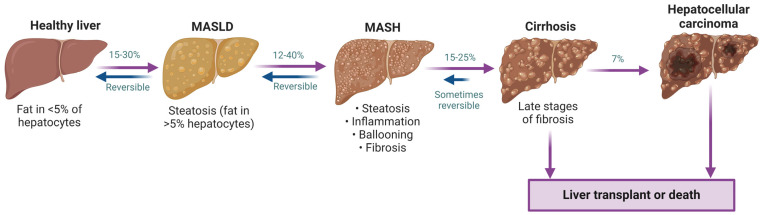
Spectrum of MASLD progression. Due to obesity, lifestyle or other factors, a healthy liver may become steatotic (TG present in >5% of hepatocytes). Steatohepatitis can develop from steatosis with the addition of inflammatory insults. Both MASLD and MASH are often reversible, while cirrhosis may partially regress with treatment. Otherwise, this can lead to hepatocellular carcinoma and may ultimately require liver transplantation [9,10].

**Figure 2 ijms-25-08731-f002:**
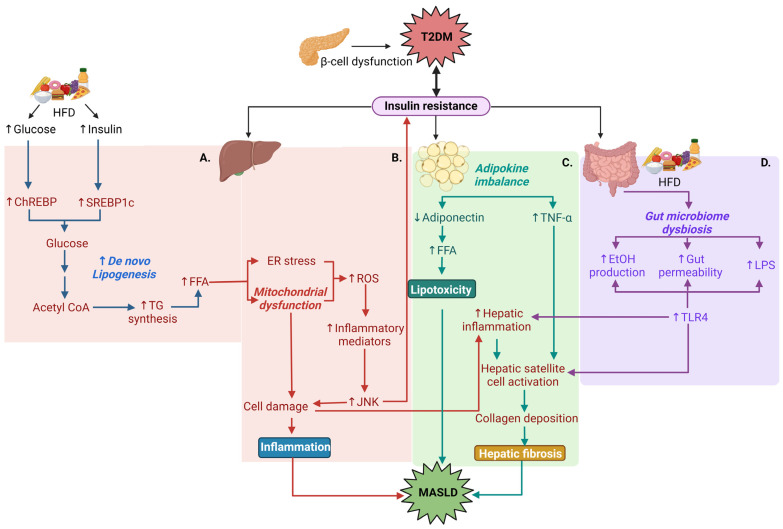
Inflammation, lipotoxicity and fibrosis contribute to the pathophysiological relationships between T2DM and MASLD. β-cell dysfunction causes IR in T2DM, followed by organ-specific pathology. (**A**) A chronic high-fat diet (HFD) can lead to IR, with increased glucose and insulin but blunted downstream receptor signaling, resulting in sustained elevated blood glucose. Glucose and insulin promote de novo lipogenesis through the activation of ChREBP and SREBP1c, respectively, to produce free fatty acids (FFAs). (**B**) Increased hepatic FFA causes mitochondrial dysfunction and ER stress followed by the rise of reactive oxygen species (ROS) and inflammatory mediators. The activation of JNK signaling by these mediators can cause hepatic cell damage and lead to inflammation; JNK signaling also decreases insulin sensitivity. (**C**) The development of IR is also related to alterations in adipokines secreted from adipose tissue. Decreased adiponectin induces lipotoxicity by increasing FFA, while increased inflammatory TNF-α drives hepatic satellite cell activation, leading to hepatic fibrosis due to excess collagen deposition. (**D**) Finally, chronic HFD affects the gut microbiota. Increased gut permeability and bacterial cellular lipopolysaccharide (LPS) or ethanol production due to dysbiosis leads to an increase in hepatic toll-like receptor 4 activation and subsequent inflammation and fibrogenesis in the liver. Red: hepatic pathways; green: adipocyte pathways; purple: intestinal pathways.

**Figure 3 ijms-25-08731-f003:**
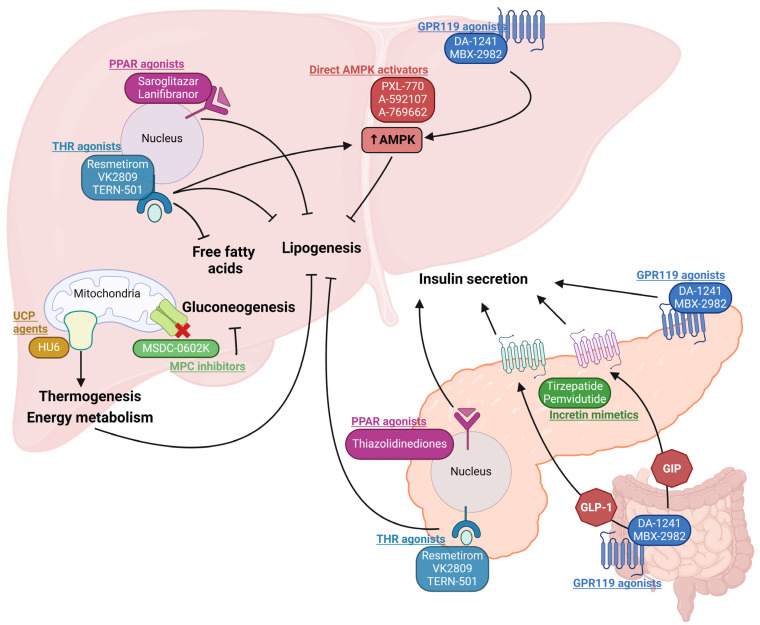
Novel treatment approaches common to T2DM and MASLD. PPAR agonists and THR agonists act to reduce hepatic free fatty acids and subsequent lipogenesis. AMPK activation is targeted via direct agonists or indirectly through THR agonists and GPR119 agonists to reduce lipogenesis. Mitochondrial targeting increases thermogenesis and reduces gluconeogenesis to ultimately reduce lipogenesis and steatosis in the liver. In the pancreas, PPAR and THR agonists act to increase insulin secretion and reduce hepatic lipogenesis. In the distal small intestine, GRP119 agonists as well as incretin mimetics potently stimulate insulin release and promote weight loss.

**Figure 4 ijms-25-08731-f004:**
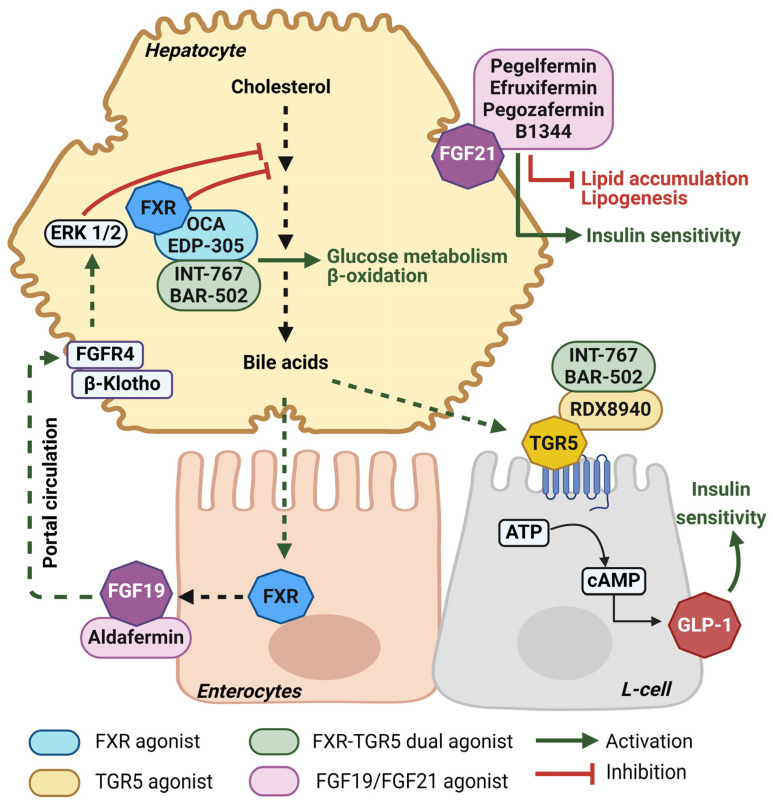
Bile-acid-based treatments for MASLD. Bile acids are synthesized from cholesterol and are released into the duodenum post-prandially to aid in nutrient and fat digestion. Bile acids also activate the receptors FXR and TGR5 to regulate metabolism. In enterocytes, the activation of FXR by bile acids or agonists induces FGF19 release which suppresses bile acid synthesis and has been shown to reduce steatosis. In hepatocytes, the activation of FXR suppresses bile acid synthesis and promotes glucose homeostasis and β-oxidation. Also in the liver, FGF21 promotes insulin sensitivity and reduces lipogenesis. In intestinal L-cells, the activation of TGR5 promotes insulin sensitivity through the release of GLP-1.

**Table 1 ijms-25-08731-t001:** Current treatment approaches for T2DM.

Category	Class	Examples
Insulin	Rapid-acting (3–5 h)	Lispro, Aspart
	Rapid-acting inhaled (3 h)	Human inhaled insulin
	Short-acting (5–8 h)	Regular human insulin
	Intermediate-acting (14–24 h)	Neutral protamine hagedorn (NPH)
	Long-acting (~24 h)	Glargine (U100), Detemir (U100)
	Ultra-long acting (>36 h)	Glargine (U300), Degludec
Insulin secretagogue	Sulfonylurea	Glipizide, Glyburide, Glimepiride
	Meglitinide	Repaglinide, Nateglinide
	Glucagon-like peptide-1 (GLP-1) receptor agonist	Semaglutide, Exenatide, Liraglutide, Dulaglutide
	Dipeptidyl peptidase-4 (DPP-4) inhibitor	Sitagliptin, Lingagliptin
Oral hypoglycemic agents	Biguanides	Metformin
	Thiazolidinediones (TZD)	Rosiglitazone, Pioglitazone
	Sodium–glucose cotransporter 2 (SGLT2) inhibitors	Empagliflozin, Dapagliflozin, Canagliflozin
	Alpha-glucosidase inhibitors	Acarbose, Miglitol

**Table 2 ijms-25-08731-t002:** Established drugs with possible MASLD indications.

Class	Example	Off-Label Indication for MASLD	Trial
Biguanide	Metformin	Reduced serum aminotransferases in patients with MASH	NCT00063232
GLP-1 agonist	Liraglutide	Resolution of MASH with no worsening of fibrosis	NCT01237119
SGLT2 inhibitor	Empagliflozin	Reduced steatosis in patients with T2DM and MASLD	NCT02686476
Thiazolidinedione	Pioglitazone	Reduced steatosis in patients with MASH and without T2DM	NCT00063622
Anti-lipidemic	Ezetimibe	Improved insulin resistance in patients with MASLD	N/A

**Table 3 ijms-25-08731-t003:** Additional novel treatment targets for MASLD.

Class	Example	Trial
Acetyl-CoA Carboxylase Inhibitor	Fircostat	NCT02856555
		NCT03987074
		NCT04971785
		NCT03449446
	PF-05221304	NCT03248882
		NCT03776175
Fatty Acid Synthase Inhibitor	Denifanstat	NCT04906421
Stearoyl-CoA Desaturase Inhibitor	ION224	NCT04932512
	PF-06865571	NCT03513588
Galectin-3 Inhibitor	Belapectin	NCT04365868

## Data Availability

No new data were created or analyzed in this study. Data sharing is not applicable to this article.

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
