# Peer review of "Pathophysiological Relationship between Type 2 Diabetes Mellitus and Metabolic Dysfunction-Associated Steatotic Liver Disease: Novel Therapeutic Approaches"

_ijms, 2024, doi:10.3390/ijms25168731_

Round 1

Reviewer 1 Report

Comments and Suggestions for Authors

Interesting and overall well-balanced review tapping into the intersection of T2D and MASLD pathophysiology and new therapeutic options. 

General comments

1. In spite of the title and what is specifically referred as the main aims of the review both in the abstract and the introduction, there is not much focus on the joint pathophysiology of T2D and MASLD, particularly systemic mechanisms are largely neglected. While I do not see much interest, for instance, in naming or succintly describing the types of treatments available for T2D and MASLD if they are not related to the pathophysiological pathways that are reviewed in this article, I believe going into more detail with the latter and then expading on the evidence available regarding metabolic control in general and then T2D and MASLD-specific endpoints for the tested drugs, either alone or in combination, would provide a unique value to the present work. 

2. In the same vein, while there are several references to the epidemiological interrelation between T2D and MASLD, the authors do not specifically tackle the interactions regarding the natural history (e.g., how MASLD affects the likelihood of developing T2D and developing other T2D-related complications and vice versa, how does poor metabolic control in T2D impact the progression of MASLD and the risk of liver-related events). I believe this point should be strenghtened. 

Specific comments

1. Abstract. "These two disorders may act synergistically, owing to increased lipotoxicity and inflammation within the liver". These are not the only mechanisms linking the two conditions. I suggest to rephrase or remove. 

2. Keyword. I suggest to include: MASLD or metabolic dysfunction-associated steatotic liver disease. 

3. Introduction. "fatty liver disease". As authors have used the new nomenclature (i.e., MASLD) across the manuscript, they should also comply with avoiding the term "fatty" as part of the proposal. Replace it by "steatotic liver disease" across the text. 

4. Introduction and elsewhere. Same thing with metabolic associated fatty liver disease and MAFLD instead of MASLD. 

5. Section 1.1. "Diabetes mellitus, or hyperglycemia". Please, clarify, as DM is not synonymous with hyperglycemia.  

6. Section 1.1. MASLD and T2D cannot be considered "health emergencies" (i.e., rapidly spreading and requiring immediate and coordinated cross-country action as in the case of COVID-19) but "global public health issues". 

7. Section 1.2. "This distressing condition is rising synchronously with the global epidemics of obesity and insulin resistance[3]." I'd try to use insulin resistance and T2D consistently across the text (not indistinctly). 

8. Figure 1. Please include the sources of the numbers used in the figure as references in the footnote. 

9. Section 1.2/Figure 1. "Irreversible cirrhosis". Available evidence suggest that a substantial proportion of patients with F4/compensated cirrhosis do have certain room for "cirrhosis regression". 

10. Section 1.2., line 71. "Insulin resistance" acronym (IR) already defined earlier in the text. 

11. Line 72. "Approximately 80% of patients with MASLD are obese, and, more precisely, the rates of obesity in the individuals with MASLD and MASH are 51.3% and 81.8% respectively[6]." This sentence does not make sense, as MASH patients only represent around 20% of total MASLD patients. 

12. "The very recently approved (March 2024) generic Resmetirom is the only treatment option for liver scarring due to MASH". "Generic" to be removed. It is not only for MASH with liver fibrosis, but the only treatment available approved by a regulatory agency for MASLD and MASH in general (with the exception of saroglitazar in India). 

13. Section 1.3. "There is an intricate relationship between obesity, T2DM, and MASLD." At this point, this has already been said several times in the text. 

14. Section 1.3. "Obesity often leads to the development of T2DM, and these complications are significant". What complications? Please, clarify. 

15. Section 1.3. ". The global pooled prevalence of MASLD among patients with T2DM was 65.33%, which increased from 55.86% in 1990-2004[10]. Insulin resistance is a major contributing phenotype to the development of MASH". Epidemiology has already been covered in the introduction. Section 1.3. is supposed to focus on pathophysiology. 

16. "Insulin resistance decreases glycogen synthesis and increases hepatic gluconeogenesis and glycolysis, which provides substrates for de novo lipogenesis and thereby results in hepatic steatosis[13](Figure 3a)". I assume here authors are referring to Figure 2 (which by the way needs better resolution, I have not been able to analyse the detail). 

17. Section 1.3. needs to clarify the distinction between liver lipotoxicity and the mechanisms linking adipose tissue inflammation and insulin resistance with liver fibrosis and adipose tissue fibrosis. 

18. Line 131. "activate FXR". Define acronym at first use. 

19. Line 174. "Additional trials (NCT04193436, NCT03256526) were completed to examine the efficacy of PF-06835919 in reducing steatosis and metabolic dysfunction associated with MASLD and T2DM." In this case and others, I'd suggest to either include a reference with a publication linked to the RCT or, in case such data are yet not published, include a reference that contains the link to the clinicaltrials.gov website for each trial. 

20. Lines 208-9. "2024 the first-ever and only MASH treatment drug, Resmetirom (under the brand) name Rezdiffra by Madrigal Pharmaceuticals) was approved by the FDA to treat liver fibrosis." Again, it is not approved to treat only liver fibrosis but also steatohepatitis (although it is true that it is indicated only in patients that have significant liver fibrosis). 

21. Lines 215-6. "only about 25% of patients receiving resmetirom responded with improvement of fibrosis without worsening of MASLD[7]". "MASH" instead of MASLD in this case. 

22. Lines 221-228 and table 2. "One such approach is through the investigation of established drugs with off-label benefits for MASLD. Many of these drugs are already approved for hypoglycemia, obesity, or hypertriglyceridemia because of their pathophysiological correlation with MASLD progression, and select drugs are listed in Table 2." First, it is not accurate to say that such drugs have been approved for indications other than MASLD because of their benefits in MASLD. Second, "hypoglycemia" does not apply here. Third, I would not, in any case refer to such drugs are approved drugs for MASLD. Authors may consider referring to certain international guidelines that have include such drugs in their recommendations in spite of the lack of evidence coming from Phase 3 trials or meta-analyses (e.g. Cusi et al. Endocrinol Pract 2022). 

23. The discussion should be expanded and tap into the most interesting pathophysiological interactions in T2D-MASLD and how this interrelates with the ongoing and potentially future drug development strategies. 

Comments on the Quality of English Language

.

Author Response

REVIEWER 1

General comments
In spite of the title and what is specifically referred as the main aims of the review both in the abstract and the introduction, there is not much focus on the joint pathophysiology of T2D and MASLD, particularly systemic mechanisms are largely neglected. While I do not see much interest, for instance, in naming or succintly describing the types of treatments available for T2D and MASLD if they are not related to the pathophysiological pathways that are reviewed in this article, I believe going into more detail with the latter and then expading on the evidence available regarding metabolic control in general and then T2D and MASLD-specific endpoints for the tested drugs, either alone or in combination, would provide a unique value to the present work.
Response: We expanded on Section 4, which highlights novel drugs for T2DM and MASLD. We include more detail regarding the mechanisms of action of GPR119 and PPAR activators, as well as THR agonists and MPC inhibitors

In the same vein, while there are several references to the epidemiological interrelation between T2D and MASLD, the authors do not specifically tackle the interactions regarding the natural history (e.g., how MASLD affects the likelihood of developing T2D and developing other T2D-related complications and vice versa, how does poor metabolic control in T2D impact the progression of MASLD and the risk of liver-related events). I believe this point should be strengthened.

Response: We included several additional studies/references that demonstrate further connections between T2DM and MASLD in the introduction and throughout the paper, including connections between HbA1c and fibrosis development, and risks of comorbidity between the two conditions as the reviewer noted above.

Specific comments
Abstract. "These two disorders may act synergistically, owing to increased lipotoxicity and inflammation within the liver". These are not the only mechanisms linking the two conditions. I suggest to rephrase or remove.

Response: We agree with this critique and did not intentionally imply that these are the sole mechanisms of cause. We rephrased the line to “These two disorders may act synergistically, in part due to increased lipotoxicity and inflammation within the liver, among other causes.”

Keyword. I suggest to include: MASLD or metabolic dysfunction-associated steatotic liver disease.
Response: We added MASLD as a keyword.

Introduction. "fatty liver disease". As authors have used the new nomenclature (i.e., MASLD) across the manuscript, they should also comply with avoiding the term "fatty" as part of the proposal. Replace it by "steatotic liver disease" across the text.
Response: We corrected all instances of “fatty liver disease” to “steatotic liver disease”.

Introduction and elsewhere. Same thing with metabolic associated fatty liver disease and MAFLD instead of MASLD.
Response: We corrected this oversight to MASLD.

Section 1.1. "Diabetes mellitus, or hyperglycemia". Please, clarify, as DM is not synonymous with hyperglycemia.
Response: We removed the term “hyperglycemia” from this section.

Section 1.1. MASLD and T2D cannot be considered "health emergencies" (i.e., rapidly spreading and requiring immediate and coordinated cross-country action as in the case of COVID-19) but "global public health issues".
Response: We rephrased “health emergencies” to “global public health issues”.

Section 1.2. "This distressing condition is rising synchronously with the global epidemics of obesity and insulin resistance[3]." I'd try to use insulin resistance and T2D consistently across the text (not indistinctly).
Response: We changed “insulin resistance” to T2DM across the text where appropriate.

Figure 1. Please include the sources of the numbers used in the figure as references in the footnote.
Response: We added references for Figure 1 in the legend.

Section 1.2/Figure 1. "Irreversible cirrhosis". Available evidence suggest that a substantial proportion of patients with F4/compensated cirrhosis do have certain room for "cirrhosis regression".
Response: We omitted the word “irreversible” before cirrhosis in section 1.2 and changed Figure 1 accordingly, and we also added a reference for support.

Section 1.2., line 71. "Insulin resistance" acronym (IR) already defined earlier in the text.
Response: We changed all instances of “Insulin resistance” to “IR” afterwards.

Line 72. "Approximately 80% of patients with MASLD are obese, and, more precisely, the rates of obesity in the individuals with MASLD and MASH are 51.3% and 81.8% respectively[6]." This sentence does not make sense, as MASH patients only represent around 20% of total MASLD patients.
Response: We modified this statement for clarity, which now reads “the rates of obesity in MASLD and MASH patients are 51.3% and 81.6%, respectively.”

"The very recently approved (March 2024) generic Resmetirom is the only treatment option for liver scarring due to MASH". "Generic" to be removed. It is not only for MASH with liver fibrosis, but the only treatment available approved by a regulatory agency for MASLD and MASH in general (with the exception of saroglitazar in India).
Response: We changed the word “generic” to “drug” and rephrased the line accordingly.

Section 1.3. "There is an intricate relationship between obesity, T2DM, and MASLD." At this point, this has already been said several times in the text.
Response: We removed this sentence.

Section 1.3. "Obesity often leads to the development of T2DM, and these complications are significant". What complications? Please, clarify.
Response: We clarified this point to read “conditions” instead of “complications”.

Section 1.3. ". The global pooled prevalence of MASLD among patients with T2DM was 65.33%, which increased from 55.86% in 1990-2004[10]. Insulin resistance is a major contributing phenotype to the development of MASH". Epidemiology has already been covered in the introduction. Section 1.3. is supposed to focus on pathophysiology.
Response: We agree with this critique; this line was moved to section 1.2 where epidemiology is discussed.

"Insulin resistance decreases glycogen synthesis and increases hepatic gluconeogenesis and glycolysis, which provides substrates for de novo lipogenesis and thereby results in hepatic steatosis[13](Figure 3a)". I assume here authors are referring to Figure 2 (which by the way needs better resolution, I have not been able to analyse the detail).
Response: We corrected the figure number in the text. The figures were originally uploaded as 600 dpi jpeg files that may have been degraded via the uploading or processing steps; we reloaded all figure files as a precaution.

Section 1.3. needs to clarify the distinction between liver lipotoxicity and the mechanisms linking adipose tissue inflammation and insulin resistance with liver fibrosis and adipose tissue fibrosis.
Response: With supporting reference, here we have added a new paragraph where we discussed how decreased adiponectin level due to IR causes excess free fatty acid efflux in hepatocytes, following lipotoxicity due to imbalanced mitochondrial β-oxidation. This lipotoxicity further provokes inflammatory mediators, resulting in fibrosis.

Line 131. "activate FXR". Define acronym at first use.
Response: We corrected this oversight and defined FXR (Farnesoid X receptor).

Line 174. "Additional trials (NCT04193436, NCT03256526) were completed to examine the efficacy of PF-06835919 in reducing steatosis and metabolic dysfunction associated with MASLD and T2DM." In this case and others, I'd suggest to either include a reference with a publication linked to the RCT or, in case such data are yet not published, include a reference that contains the link to the clinicaltrials.gov website for each trial.
Response: We agree that linking information about clinical trials is necessary; therefore, we provided published references where available and wealso provided a link to each clinical trial regardless of publication status.

Lines 208-9. "2024 the first-ever and only MASH treatment drug, Resmetirom (under the brand) name Rezdiffra by Madrigal Pharmaceuticals) was approved by the FDA to treat liver fibrosis." Again, it is not approved to treat only liver fibrosis but also steatohepatitis (although it is true that it is indicated only in patients that have significant liver fibrosis).
Response: We corrected this statement to read “liver fibrosis and also steatohepatitis”.

Lines 215-6. "only about 25% of patients receiving resmetirom responded with improvement of fibrosis without worsening of MASLD[7]". "MASH" instead of MASLD in this case.
Response: We corrected this oversight.

Lines 221-228 and table 2. "One such approach is through the investigation of established drugs with off-label benefits for MASLD. Many of these drugs are already approved for hypoglycemia, obesity, or hypertriglyceridemia because of their pathophysiological correlation with MASLD progression, and select drugs are listed in Table 2." First, it is not accurate to say that such drugs have been approved for indications other than MASLD because of their benefits in MASLD. Second, "hypoglycemia" does not apply here. Third, I would not, in any case refer to such drugs are approved drugs for MASLD. Authors may consider referring to certain international guidelines that have include such drugs in their recommendations in spite of the lack of evidence coming from Phase 3 trials or meta-analyses (e.g. Cusi et al. Endocrinol Pract 2022).
Response: We realized this was a confusing sentence and we intended to imply correlation between the diseases listed and MASLD, not drugs and MASLD. Therefore, we modified this sentence to: “Clinical trials are being conducted to determine if these drugs can be utilized for the management of MASLD, in part due to the pathophysiological correlation of these conditions with MASLD progression”. We also corrected the typo “hypoglycemia” to “hyperglycemia” and rephrased “Off-label indication” to “possible indication”. Finally, we added the above reference as suggested by the reviewer.

The discussion should be expanded and tap into the most interesting pathophysiological interactions in T2D-MASLD and how this interrelates with the ongoing and potentially future drug development strategies.
Response: We expanded the discussion with references from primary articles to highlight the most promising T2DM-MASLD pathophysiological drug targets from our standpoint, which are FXR-TGR5 dual agonists. We also refer to patient-to-patient differential pathophysiological alterations, and discuss that both therapeutic and management approaches should be personalized accordingly to treat these two bidirectional conditions.

Reviewer 2 Report

Comments and Suggestions for Authors

The facts of existence the new therapy for patients with MASLD and T2DM. This review make a very clear explanation of the pathophisiology of these drugs. This review aimis to discuss the pathogenic interactions between T2DM and MASLD and high-light novel therapeutic drugs.

There are no conclusions describe in this review, and at the discussion section there are very few data from the literature. They must improve this section, and describe the efficacy of Resmetirom in these patients.

I suggest to improve the section of discussions. I think that additional information is needed on this topic, especially regarding GLP-1 analogues and SGLT-2 inhibitors. It also requires the presentation of a conclusion

Author Response

REVIEWER 2

The facts of existence the new therapy for patients with MASLD and T2DM. This review make a
very clear explanation of the pathophisiology of these drugs. This review aimis to discuss the
pathogenic interactions between T2DM and MASLD and high-light novel therapeutic drugs.
There are no conclusions describe in this review, and at the discussion section there are very few
data from the literature. They must improve this section, and describe the efficacy of Resmetirom
in these patients.

I suggest to improve the section of discussions. I think that additional information is needed on
this topic, especially regarding GLP-1 analogues and SGLT-2 inhibitors. It also requires the
presentation of a conclusion

Response: We added several references and additional information regarding Resmetirom as
suggested. We also agree that SGLT2 inhibitors are an important drug class demonstrating
favorable effects for MASLD, so we added a new subsection to section 4.0 Novel Common
Approaches. Finally, we added more detail related to incretins and GLP-1.

Reviewer 3 Report

Comments and Suggestions for Authors

Journal: IJMS    Manuscript ID: ijms-3103919

Authors:  Shifat-E Ferdous and Jessica M. Ferrell.

Title: “Pathophysiological Relationship between Type 2 Diabetes Mellitus and Metabolic Dysfunction-Associated Steatotic Liver Disease and Their Novel Therapeutic Approaches”

The authors in the present review discuss the growing global concern of Type 2 diabetes mellitus (T2DM) and metabolic dysfunction-associated steatotic liver disease (MASLD), emphasizing the interplay between T2DM and MASLD. The review addresses pathophysiological mechanisms and highlights new therapeutic targets and ongoing clinical trials for treating these interconnected disorders. The following points merit consideration:

Comments:

1.     Several sections (e.g., 1.1, 1.2, 1.3) contain epidemiological information. I suggest creating a separate paragraph with the epidemiological data, highlighting the figures for each disease's burden as well as the common epidemiology of their coexistence, including relevant statistics.

2.     In the introduction, it would be helpful to emphasize the importance of this review and what it adds to the literature, in addition to other existing similar reviews.

3.     Given the focus of this paper on the pathophysiological relationship between T2DM and MASLD, I would suggest expanding more on the underlying pathophysiological mechanisms and, where available, highlight the interconnection between these disorders. This will help the reader better understand each pathophysiological mechanism described. For example, the statement "obesity and hyperlipidemia due to imbalance in adipokines from adipose tissue that results in lipotoxicity and collagen deposition" is concise. Expanding further would help the reader understand this mechanism and how the two diseases are pathophysiologically connected. Likewise, this approach should be applied to other mechanisms described as appropriate.

4.     The section "Treatment approaches for T2DM" includes Table 1, which summarizes the current treatment approaches for T2DM and then continues with the novel treatment strategies for T2DM. Likewise, for MASLD. Perhaps a different structure grouping current approaches that are common for T2DM and MASLD, novel approaches that are common for T2DM and MASLD, and the novel approaches unique for each condition separately would facilitate reading and highlight the interconnection of the two diseases further.

5.     According to the authors, “Both MASLD and T2DM are induced by hypothyroidism, which typically involves increased secretion of TSH but reduced circulating T3 and T4. Figure 7 demonstrates how THR agonists can target hypothyroidism-induced MASLD and T2DM.” Please add the relevant reference. Also, it would be informative to add relevant comments in the Pathophysiological relationships between T2DM and MASLD section. Moreover, do the authors mean “Figure 4” instead of Figure 7? In the PDF I have up to Figure 4. Please clarify this point.

6.     I would suggest avoiding using the adjective "diabetic" to characterize individuals. For instance, consider using phrases such as "individuals/patients/participants with diabetes," "patients living with diabetes," etc.

Author Response

REVIEWER 3
Several sections (e.g., 1.1, 1.2, 1.3) contain epidemiological information. I suggest creating a separate paragraph with the epidemiological data, highlighting the figures for each disease's burden as well as the common epidemiology of their coexistence, including relevant statistics.
Response: We moved most of the epidemiological information to a single paragraph.

In the introduction, it would be helpful to emphasize the importance of this review and what it adds to the literature, in addition to other existing similar reviews.
Response: We added the following statement to the Introduction: “Our aim is to highlight the substantial overlap between these two conditions, with the hope it may prepare healthcare providers to pre-emptively screen and monitor patients who are vulnerable to these comorbidities.”

Given the focus of this paper on the pathophysiological relationship between T2DM and MASLD, I would suggest expanding more on the underlying pathophysiological mechanisms and, where available, highlight the interconnection between these disorders. This will help the reader better understand each pathophysiological mechanism described. For example, the statement "obesity and hyperlipidemia due to imbalance in adipokines from adipose tissue that results in lipotoxicity and collagen deposition" is concise. Expanding further would help the reader understand this mechanism and how the two diseases are pathophysiologically connected. Likewise, this approach should be applied to other mechanisms described as appropriate.
Response: As noted by the other Reviewers, we expanded on the sections that highlight the pathological connections between T2DM and MASLD.

The section "Treatment approaches for T2DM" includes Table 1, which summarizes the current treatment approaches for T2DM and then continues with the novel treatment strategies for T2DM. Likewise, for MASLD. Perhaps a different structure grouping current approaches that are common for T2DM and MASLD, novel approaches that are common for T2DM and MASLD, and the novel approaches unique for each condition separately would facilitate reading and highlight the interconnection of the two diseases further.
Response: We reverse the presentation order of these topics.

According to the authors, “Both MASLD and T2DM are induced by hypothyroidism, which typically involves increased secretion of TSH but reduced circulating T3 and T4. Figure 7 demonstrates how THR agonists can target hypothyroidism-induced MASLD and T2DM.” Please add the relevant reference. Also, it would be informative to add relevant comments in the Pathophysiological relationships between T2DM and MASLD section. Moreover, do the authors mean “Figure 4” instead of Figure 7? In the PDF I have up to Figure 4. Please clarify this point.
Response: We added the correct reference and corrected the Figure. We also added information regarding the pathophysiological connections between T2DM and MASLD as noted.

I would suggest avoiding using the adjective "diabetic" to characterize individuals. For instance, consider using phrases such as "individuals/patients/participants with diabetes," "patients living with diabetes," etc.
Response: We corrected these instances to patient-forward language throughout.

Round 2

Reviewer 1 Report

Comments and Suggestions for Authors

The authors have satisfactorily provided answers and/or made changes to address my comments. 

Reviewer 3 Report

Comments and Suggestions for Authors

Journal: IJMS    Manuscript ID: ijms-3103919 (Revised version)

Authors:  Shifat-E Ferdous and Jessica M. Ferrell.

Title: “Pathophysiological Relationship between Type 2 Diabetes Mellitus and Metabolic Dysfunction-Associated Steatotic Liver Disease and Their Novel Therapeutic Approaches”

The authors of the present review article have satisfactorily responded to my comments and suggestions and made the necessary changes to the paper. There are no further comments.